# “Anees Has Measles”: Storytelling and Singing to Enhance MMR Vaccination in Child Care Centers Amid Religious Hesitancy

**DOI:** 10.3390/vaccines12070819

**Published:** 2024-07-22

**Authors:** Therdpong Thongseiratch, Puttichart Khantee, Naphat Jaroenmark, Napatsaree Nuttapasit, Nithida Thonglua

**Affiliations:** 1Child Development Unit, Department of Pediatrics, Faculty of Medicine, Prince of Songkla University, Songkhla 90110, Thailand; natpat.nay@hotmail.com (N.J.); napatsaree.nut@gmail.com (N.N.); thonglua31@gmail.com (N.T.); 2Infectious Disease Unit, Department of Pediatrics, Faculty of Medicine, Prince of Songkla University, Songkhla 90110, Thailand

**Keywords:** MMR vaccination, cultural adaptation, storytelling, singing, health education, vaccine hesitancy

## Abstract

This study explores the enhancement of MMR vaccination rates in the deep south of Thailand through a culturally tailored intervention that incorporates storytelling and singing, aligning with local cultural and religious values. The “Anees has Measles” intervention, developed with input from community stakeholders including health workers, religious leaders, and parents, featured traditional melodies in its singing activities. The intervention comprised two main components: storytelling sessions and singing activities, both utilizing culturally resonant content and formats to effectively engage the community. Conducted from December 2019 to March 2020 across eight districts in five provinces, the study targeted children aged 18 months to 5 years in government child care centers. Results indicated a substantial increase in MMR vaccination coverage from 44.3% to 72.0% twelve months post-intervention, and further to 77.0% at 48 months post-intervention, representing a significant and sustained improvement (*p* < 0.001). This marked and continuous increase demonstrates the effectiveness of culturally adapted health education in areas with significant vaccine hesitancy. The findings highlight the importance of integrating culturally and religiously sensitive methods into public health strategies, significantly enhancing vaccine acceptance and coverage in diverse and conservative settings. This approach suggests a broader applicability for similar interventions in comparable contexts globally.

## 1. Introduction

Measles is a highly contagious viral disease that is spread through the air by respiratory droplets when infected individuals cough or sneeze [1]. If left untreated, measles can lead to serious complications such as encephalitis, pneumonia, and even death [2]. The measles vaccine, often administered as part of the combined measles, mumps, and rubella (MMR) vaccine, is a highly effective measure to prevent the spread of the disease [3]. The World Health Organization (WHO) recommends that children receive two doses of the vaccine for maximum protection [4,5]. The vaccine has been instrumental in reducing measles cases globally, and high vaccination coverage is essential for achieving herd immunity and preventing outbreaks [6].

In recent years, measles outbreaks have increasingly occurred in communities with low vaccination rates [7,8,9]. Unvaccinated children are at a higher risk of contracting measles and contributing to the spread of the virus [10]. These outbreaks are particularly concerning in regions where healthcare access is limited, and the capacity to manage large-scale measles cases is inadequate. The resurgence of measles in certain areas high-lights the importance of maintaining high vaccination coverage to prevent the disease’s spread [11,12,13].

Religious and cultural beliefs contribute to vaccine hesitancy [14]. For instance, in parts of Southern Thailand, there are misconceptions about the MMR vaccine not being halal, leading to lower vaccination rates among Muslim populations [15,16,17]. The perception that vaccines contain ingredients forbidden in Islam has fueled skepticism and fear, resulting in decreased uptake of essential immunizations like the measles vaccine [18,19].

To address the challenges posed by MMR vaccine hesitancy, various interventions have been implemented [20]. These include educational campaigns to dispel myths about vaccines, engagement with religious and community leaders to endorse vaccination, and culturally sensitive health communication strategies [21,22,23,24]. Many factors influence pa-rental decisions to vaccinate, and parents who actively decline immunization often believe that vaccines are unsafe and ineffective, that the diseases they are meant to prevent are mild and uncommon, and may mistrust their health professionals [25]. Nevertheless, such attitudes could shift during an outbreak [26,27]. To date, there has been limited research assessing the uptake of the MMR vaccine following measles outbreaks in communities with historically low vaccination rates. This information is crucial for managing future outbreaks in similar contexts.

Since 2000, measles cases in Thailand have generally trended downwards, but the deep southern provinces have experienced recurring epidemics every 3–5 years, leading to disability and mortality. These provinces, where Muslims are the majority, reported the highest incidence rates in the country, with 126 cases per 100,000 population, and notably low MMR vaccine coverage of around 70% from 2014 to 2018 [28,29,30]. The region’s overall childhood vaccine coverage is among the lowest in Thailand, compounded by them having the highest frequency of measles outbreaks. In November 2018, more than 20 cases of measles-related deaths were reported for that year’s outbreak, highlighting the severe impact of these epidemics [28,29,30,31,32]. In this area, vaccine acceptance is significantly influenced by Islamic practices and religious leaders, contrasting with other Thai communities, predominantly Buddhist, who generally perceive vaccines as protective against disease [33]. This pattern is similar to that observed in Malaysian Muslim communities, where vaccine hesitancy is also prevalent [34].

In this study, we explored the impact of a measles outbreak and a targeted two-pronged immunization campaign on MMR vaccine uptake at local child care centers in a region with notable religious hesitancy. In Southern Thailand, initiatives employing design thinking methods have been used to empathize with community concerns and develop new strategies to boost MMR vaccination. These efforts focus on creating culturally resonant narratives and imagery to enhance health literacy and promote vaccine acceptance among child care center staff and parents. The objectives of the study were to (a) describe and compare the uptake of MMR at the local child care centers with that of the wider regional and national levels, (b) assess trends in immunization coverage and evaluate any increases in relation to the timing of the outbreak and the immunization campaign, and (c) gather quantitative and qualitative feedback from child care center staff regarding the impact of the immunization campaign.

## 2. Materials and Methods

### 2.1. Study Population

The study population included parents, child care center staff, and religious counselors associated with children aged 18 months to 5 years in subdistrict local government child care centers across 8 districts with recorded measles mortality, located in 5 provinces in the deep south of Thailand. The data collection period spanned from 1 December 2019 to March 2020. During this time, an immunization campaign was implemented, which included distributing a culturally adapted storybook and organizing measles prevention activities such as storytelling and singing at the child care centers. These districts were selected based on their historically low uptake of the MMR vaccine and recent reports of measles outbreaks, targeting areas where the intervention could have the most significant impact.

### 2.2. Storytelling and Singing Intervention

The intervention included two main components designed to increase MMR vaccination rates among the target population: the distribution of a specially designed storybook and the conducting of related singing activities.

#### Storybook Development

The storybook titled “Anees has Measles” (Figure 1) was crafted using a design thinking approach to deeply understand and empathize with the community’s concerns about MMR vaccination. This creative process involved various stakeholders from the community, including parents, health workers, and religious leaders, who shared their insights on cultural nuances and common hesitancies surrounding vaccines. Their contributions ensured the content was culturally relevant and sensitive to the audience’s beliefs and practices.

The storybook aims to engage both children and their parents with its culturally resonant narratives and appealing imagery, making the topic of vaccination approachable and understandable. It consists of the following components:(1)The Story of Anees: A narrative about a girl named Anees who contracts measles, illustrating the severity of the disease. It underscores the message that vaccines can be deemed halal when necessary to save lives, aligning with religious considerations.(2)Insights from the Muslim Organization of Thailand: This section highlights the importance of vaccinations and reassures that they can be considered halal, addressing a significant cultural concern within the community.(3)Educational Q&A on MMR Vaccine: Eight carefully crafted questions and answers designed to educate the community about the MMR vaccine, addressing common doubts and misinformation.(4)Vaccine EPI Schedule Table of Thailand: An informative table presenting the official vaccination schedule, helping parents understand when their children should receive their shots.(5)Vaccine Song Lyrics: Lyrics of a song related to the MMR vaccine, designed to be catchy and memorable, which helps reinforce the information about vaccination in a fun and engaging way.

Singing Activity: Alongside the storybook, a singing activity was integrated into the intervention. This activity utilized traditional Muslim melodies to reinforce the messages in the storybook, emphasizing the benefits of vaccination and the risks of measles. The singing sessions were designed to be interactive, involving both children and their parents, and thereby fostering a community-based learning environment that supports behavioral change towards the utilization of health practices. The complete videos of the songs performed in the intervention can be accessed via https://photos.app.goo.gl/GYm4Ft5pBEY2Fbjw5 (accessed on 1 July 2024). All teachers, medical students, and children featured in these videos have provided their consent.

### 2.3. Targeted Vaccination Categories

This study focused exclusively on a catch-up vaccination approach:

C1 (Catch-up Vaccination): Targeted children aged 18 months to 5 years who had not previously received any MMR doses. These children were offered their first dose of MMR, with the recommendation to receive a second dose at least one month later.

C2 (Catch-up Vaccination): Consisted of children who had already received one dose of MMR. These children were provided a second dose to complete their immunization against measles.

### 2.4. Data Sources and Collection

Privacy, confidentiality, and the rights of patients were rigorously protected throughout the study in accordance with local health information policies. Data were primarily collected from two main sources:

Local Health Data, including details extracted from the local health information system, such as gender, birth date, and dates of MMR immunizations for children.

Immunization Records, meticulously checked by child care center staff before and after the intervention; these records, combined with secondary data from the local health data system, were used to verify the immunization status of each child.

### 2.5. Data Analysis

Vaccination Uptake Analysis: MMR vaccination coverage at 12 and 48 months was assessed based on records from the local health clinics and compared with broader regional and national data. The differences in vaccination rates before and after the intervention were analyzed using the Wilcoxon signed-rank test due to the non-normal distribution of the data (as indicated by the Shapiro–Wilk test). Statistical significance was determined at a level of *p* < 0.05. Confidence intervals (CI) for the changes in vaccination rates were calculated to provide a measure of the precision of the estimated effects.

Trend Analysis in Immunization Coverage: The monthly administration of MMR doses during 2020 was tracked alongside significant outbreak events and compared with historical data from 2018 to 2019 to identify any significant changes in vaccination trends. The analysis involved comparing the pre-intervention and post-intervention periods, with statistical significance determined using appropriate non-parametric tests.

Quantitative and Qualitative Feedback: Gathered feedback from child care staff regarding the intervention’s impact, analyzed quantitatively and thematically to identify key themes affecting the program’s success.

All statistical analyses were performed using STATA/IC version 16.0, focusing on the differences in vaccination uptake before and after the intervention, and evaluating the effectiveness of the culturally tailored educational strategies implemented during the measles outbreak.

## 3. Results

The study population consisted of 983 children aged 18 months to 5 years old in a government child care center as of 1 December 2019; 49% were male.

### 3.1. Vaccination Uptake at Local Health Center, Provincial, and National Levels

Vaccination coverage was closely monitored at the local health center and compared with regional and national levels. Initially, the local health center reported an MMR2 coverage rate of 44.3% (435/983) before the intervention. This rate increased significantly to 72.0% (708/983) twelve months after the intervention and further to 77.0% (734/953) by the 48-month mark post-intervention (Table 1).

Compared to broader geographical vaccination trends, the local health center’s pre-intervention coverage was considerably lower than the regional and national averages. The deep-south region’s coverage was 50.8% initially, rising to 66.7% at 12 months but slightly declining to 65.7% at 48 months. Nationally, Thailand started at 89.7% and saw a slight increase to 90.5% at 12 months, reaching 92.5% at 48 months.

### 3.2. Statistical Analysis of Vaccination Rate Changes

The effectiveness of the “Anees has Measles” storybook intervention on MMR vaccination rates was statistically analyzed. The Shapiro–Wilk test indicated non-normal distribution of vaccination rate changes (W = 0.775, *p* < 0.001). Consequently, the Wilcoxon signed-rank test was applied, confirming a significant increase in vaccination rates from an initial 44.3% to 72.0% post-intervention, an increase of 27.77 percentage points (Z = 31,180.5, *p* < 0.001). This substantial increase underscores the intervention’s success in boosting MMR vaccination uptake in the target child care centers. These findings demonstrate notable improvements in MMR vaccination rates at the local health center, highlighting the significant impact of the targeted educational intervention. Despite remarkable progress, local rates still lag behind national averages, emphasizing the need for continued efforts to enhance vaccination coverage.

### 3.3. Quantitative Child Care Staff Feedback

Following the evaluation of vaccination coverage, an additional aspect of the study involved assessing the satisfaction levels of staff from the Subdistrict Administrative Organization and child care centers regarding the “Anees has Measles” storybook intervention. A survey was conducted to gather these insights, with the results highlighting a positive reception towards the project’s execution and outcomes.

#### 3.3.1. Satisfaction Survey Results

Program Appropriateness: A large majority of respondents felt the program’s format was highly suitable, with 28.39% rating it as mostly appropriate and 54.20% as appropriate, summing to an overall satisfaction rate of 80.21%.

Program Duration: Similarly, the duration of the program was deemed appropriate by 26.27% and very appropriate by 65.68%, yielding a combined approval of 79.86%.

Knowledge/Benefit Gained: The educational benefit of the project “Anees has Measles” was acknowledged, with 83.57% of participants reporting they gained knowledge or benefits from the program.

Increased Vaccine Awareness: The importance of measles vaccination was more widely recognized post-intervention, as indicated by an 88.39% agreement rate among the respondents.

Data Collection and Transfer: The project’s data management was praised, with 80.77% satisfaction regarding the ability to collect and transfer complete and accurate data.

The overall satisfaction with the project was noted at 83.22%, indicating strong approval of the initiative’s impact and execution.

#### 3.3.2. Book Feedback and Impact

Increased Disease Understanding: 80.51% of respondents felt their understanding of measles improved significantly.

Perceived Vaccine Importance: 81.65% recognized the increased importance of vaccination.

Confidence in Vaccination: An increased confidence in vaccinating was reported by 81.14% of participants.

Religious Context Understanding: The project facilitated a better understanding of religious stipulations regarding vaccination, with 79.62% acknowledging this aspect.

### 3.4. Qualitative Child Care Staff Feedback

In addition to the quantitative satisfaction survey, qualitative feedback was collected from participants to gain deeper insights into the experiences and perceptions surrounding the “Anees has Measles” storybook intervention. Through thematic analysis of the qualitative data, several key themes emerged that provided a nuanced understanding of the intervention’s impact and areas for improvement.

Themes from Qualitative Feedback:1.Continuity and Frequency of Programs

Participants emphasized the need for regular and ongoing interventions to maintain high levels of awareness and prevention.


*“We need this measles prevention project every year. It helps in keeping the community alert and prepared against the disease.”*


2.Community and Multi-sectoral Engagement

The importance of collaborative efforts among healthcare providers, parents, and community leaders was highlighted as crucial for the success of the program.


*“It’s crucial that everyone gets involved, not just the healthcare workers. Parents, teachers, and local leaders need to understand and support the initiative for it to be successful.”*


3.Project Duration and Reach

Respondents suggested that a longer project duration and broader reach would enhance the effectiveness of the intervention.


*“The program was great, but it felt rushed. Extending the duration and covering more areas could help reach more families and have a stronger impact.”*


4.Field Activities

Theme: The need for more interactive and on-site activities was identified to strengthen the program’s impact and community engagement.


*“Having more activities in our community would make the message stronger. People need to see and experience the benefits of vaccination firsthand.”*


5.Sustainability of Educational Efforts

Ongoing educational initiatives were deemed necessary to improve vaccine uptake and health literacy continuously.


*“Continuous education on not just measles but all vaccine-preventable diseases is needed to keep our community healthy. This should be an ongoing effort.”*


6.Barriers to Engagement

Participants noted several barriers to engagement, including logistical issues and the accessibility of educational materials.


*“The storybook was a good idea, but my work schedule didn’t allow me to participate much. Also, some parents mentioned that the text and pictures were too small for the kids to see clearly.”*


7.Enhanced Educational Materials

Feedback indicated a need for more appealing and accessible educational materials to effectively convey the message.


*“The storybook should have larger print and more colorful pictures to attract the children’s attention. Making the material more engaging will help in delivering the message effectively.”*


These themes and quotations encapsulate the multifaceted perspectives on the “Anees has Measles” storybook intervention, highlighting the program’s successes and areas for improvement. The feedback underscores the necessity for adaptable, engaging, and sustained efforts to combat vaccine hesitancy and promote public health in the community.

## 4. Discussion

The effectiveness of the “Anees has Measles” storybook intervention, complemented by singing activities, in increasing MMR vaccination rates among young children in the deep south of Thailand is a testament to the power of culturally and religiously sensitive health communication strategies. By integrating stakeholders from the community in the development process—including parents, health workers, and religious leaders—the intervention was tailored to resonate with the local cultural and religious ethos, which played a crucial role in its acceptance and success. This approach is consistent with previous studies that emphasize the importance of community involvement in health interventions to ensure cultural compatibility and increase the likelihood of positive outcomes [35,36,37].

Utilizing storytelling and singing, which are culturally familiar methods of communication, the intervention presented the vaccination message in an engaging and non-confrontational manner. This strategy proved effective in not only increasing the understanding of and trust in vaccinations but also in making the educational process enjoyable for children and their parents [38,39,40,41]. Similar approaches have been documented in other settings, where integrating health education into local cultural practices significantly improved public health initiatives’ receptiveness and effectiveness [42].

The substantial increase in vaccination rates post-intervention highlights the potential of targeted educational strategies to modify health behaviors, particularly in areas with initial vaccine hesitancy and low vaccine coverage. This aligns with findings from other regions where tailored communication strategies have successfully addressed vaccine hesitancy. Moreover, the use of specific educational tools within the storybook, such as the Q&A section and vaccination schedule, helped demystify the MMR vaccine and addressed prevalent misconceptions, a method supported by literature on health education [43,44,45]. Despite the intervention’s success, challenges such as logistical issues and limited reach were noted, suggesting the need for ongoing support and resources to ensure wider accessibility and consistency in vaccine delivery.

The study, while insightful, presents several limitations that should be considered when interpreting the results. Firstly, the absence of a control group limits the ability to conclusively attribute the observed improvements in vaccination rates solely to the intervention, despite comparisons with regional and national data. This limitation makes it challenging to definitively isolate the effect of the intervention from other external factors that may have influenced the outcomes. Secondly, the findings are based on a relatively small sample size and are confined to specific child care centers in Southern Thailand, which may not fully represent broader populations or different cultural contexts. Thirdly, the reliance on voluntary participation could introduce selection bias, as those who chose to participate might inherently have different attitudes towards vaccination compared to the general population. Additionally, we did not collect detailed information on the sociodemographic features of parents, which could provide further insights into factors affecting vaccination uptake. While the 48-month follow-up period provided valuable longitudinal data, the study could benefit from exploring other aspects such as the continuity of vaccination behavior beyond the immediate aftermath of the intervention and the long-term acceptance of vaccines in the community. These limitations underscore the need for further research with a more robust experimental design, including a control group and broader demographic sampling, to validate and expand upon these initial findings.

The findings from this study have important implications for both clinical practice and future research. Clinically, the integration of culturally appropriate educational materials and methods into routine vaccination campaigns could enhance vaccine uptake in other regions with similar cultural and religious considerations. For future research, this study highlights the need for larger-scale interventions and longer follow-up periods to verify the findings and assess the long-term sustainability of the intervention effects. It also underscores the importance of exploring different types of educational interventions that could cater to diverse cultural groups, thereby enhancing the overall effectiveness of public health campaigns aimed at increasing vaccine coverage. Further studies could also investigate the impact of combining multiple communication strategies on vaccine hesitancy to identify the most effective approaches for various populations.

## 5. Conclusions

Overall, the “Anees has Measles” intervention illustrates that culturally tailored educational strategies that employ engaging and familiar communication formats can effectively enhance vaccination uptake in communities with significant vaccine hesitancy due to religious beliefs. The positive outcomes of this study provide valuable insights for designing future public health programs that aim to bridge the gap between cultural beliefs and health behaviors, emphasizing the importance of a sustained and adaptive approach to health education and community involvement.

## Figures and Tables

**Figure 1 vaccines-12-00819-f001:**
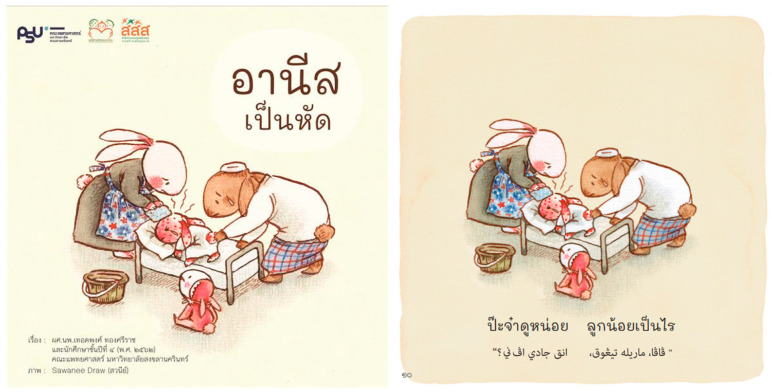
The storybook, titled “Anees has Measles”.

**Table 1 vaccines-12-00819-t001:** Vaccination coverage at the local health center, in the deep-south region, and at the national level (Thailand).

Geographical Area	MMR2 Coverage at Pre-Intervention (%)	MMR2 Coverage at 12 Months (%)	MMR2 Coverage at 48 Months (%)
Local health center	435/983 (44.3)	708/983 (72.0)	734/953 (77.0)
Deep south region	50.8	66.7	65.7
Thailand	89.7	90.5	92.5

## Data Availability

Data supporting the reported results can be reached by contacting the corresponding author. Where no new data were created, or where data are unavailable due to privacy or ethical restrictions, please see the Appendix A associated with this publication.

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
