# Peer review of "“Anees Has Measles”: Storytelling and Singing to Enhance MMR Vaccination in Child Care Centers Amid Religious Hesitancy"

_vaccines, 2024, doi:10.3390/vaccines12070819_

Round 1
Reviewer 1 Report
Comments and Suggestions for Authors
The manuscript describes the use of culturally tailored interventions, in this case storytelling and singing, to improve vaccination rates for MMR vaccine. This program was introduced in several provinces in Deep Soth Thailand, an area with low coverage because hesitancy due to religious beliefs. Vaccination rates in the monitored population increased from 43% to 77%. The results are encouraging, and the authors provide an honest assessment of the study’s limitation sin the Discussion. Vaccine hesitancy is a major impediment to achieving measles and rubella elimination and this report provides a great example of how to develop culturally tailed materials.
Author Response
Response to Reviewer 1
Reviewer Comment:
The manuscript describes the use of culturally tailored interventions, in this case storytelling and singing, to improve vaccination rates for MMR vaccine. This program was introduced in several provinces in Deep Soth Thailand, an area with low coverage because hesitancy due to religious beliefs. Vaccination rates in the monitored population increased from 43% to 77%. The results are encouraging, and the authors provide an honest assessment of the study’s limitation sin the Discussion. Vaccine hesitancy is a major impediment to achieving measles and rubella elimination and this report provides a great example of how to develop culturally tailed materials.
Author Response:
Thank you for your positive feedback on our manuscript titled "Anees has Measles: Storytelling and Singing to Enhance MMR Vaccination in Child Care Centers Amid Religious Hesitancy." We appreciate your recognition of the significant increase in vaccination rates from 43% to 77% and your acknowledgment of our honest assessment of the study’s limitations. We are pleased that you find the results encouraging and our culturally tailored approach effective. Your feedback has been invaluable, and we hope the manuscript meets your expectations as it stands.

Reviewer 2 Report
Comments and Suggestions for Authors
This manuscript deals with the results of an educational public health intervention in a Muslim community, using all parts of society and a "happy" approach with comic books and songs. Those interventions could be used to implement vaccination because both parents and society are involved and their results clearly show both early and late success in memory and vaccine-friendly behavior. Happiness and joy are powerful tools frequently forgotten by health authorities, which usually prefer to use mandatory sad orders. The article appears adequate for publication as it is presented, including the comic book in supplementary files but it could be improved by supplementary files with the complete videos of the songs performed in the intervention.
Author Response
Response to Reviewer 2
Reviewer Comment:
This manuscript deals with the results of an educational public health intervention in a Muslim community, using all parts of society and a "happy" approach with comic books and songs. Those interventions could be used to implement vaccination because both parents and society are involved and their results clearly show both early and late success in memory and vaccine-friendly behavior. Happiness and joy are powerful tools frequently forgotten by health authorities, which usually prefer to use mandatory sad orders. The article appears adequate for publication as it is presented, including the comic book in supplementary files but it could be improved by supplementary files with the complete videos of the songs performed in the intervention.
Author Response:
Thank you for your insightful feedback on our manuscript. We appreciate your recognition of the innovative use of happiness and joy through comic books and songs in our intervention. We agree that these approaches effectively foster vaccine-friendly behavior and community involvement. We are pleased to inform you that we have included the complete videos of the songs performed in the intervention in the Methods section. All teachers, medical students, and children featured in this video have provided their consent. We are glad you find the article adequate for publication and appreciate your valuable suggestions.
Revised Methods Section:
Singing Activity:
Alongside the storybook, a singing activity was integrated into the intervention. This activity utilized traditional Muslim melodies to reinforce the messages in the storybook, emphasizing the benefits of vaccination and the risks of measles. The singing sessions were designed to be interactive, involving both children and their parents, thereby fostering a community-based learning environment that supports behavioral change towards health practices. The complete videos of the songs performed in the intervention can be accessed via https://photos.app.goo.gl/GYm4Ft5pBEY2Fbjw5 . All teachers, medical students, and children featured in these videos have provided their consent.

Reviewer 3 Report
Comments and Suggestions for Authors
The article “"Anees has Measles": Storytelling and Singing to Enhance MMR Vaccination in Child Care Centers Amid Religious Hesitancy” has been reviewed . This is a community intervention to reach out hesitant parents that are not prone to vaccinate their children against measles.
The study considers how an entertainment narrative about childhood vaccination influences related attitudes and positive issue-related thinking.
A few considerations:
Introduction line 80
The aim of the study is to promote vaccine acceptance among healthcare workers and parents, I believe children do not have much to say about being vaccinated or not .
2. Materials and Methods
2.1 Study Population: The study population consisted of all children aged 18 months to 5 years only? The intervention seems to focus on children but they really have no power to determine their immunization . Target population should be their parents, child care center’s staff , religious counselors etc, that should be instructed through storytelling to their children. If a district has a low uptake for MMR it isn’t because children refuse to get vaccinated.
In the context of an outbreak people become more concerned about the disease and therefore are more prone to protect their children. In fact the survey is carried out among these adult populations, so this should be metioned as study population in methods.
-What statistical analysis should be in the Methods section. Levels of significance , CI?
3. Results
Statistical assessment should include age, and sociodemographic features of parents that can affect uptake when compare to regional and national level
· coverage rate of 44.3% (435/983) before the intervention…. Was this also before the outbreak? How many children were affected in the outbreak? Were there any deaths?
Author Response
Response to Reviewer 3
Reviewer 3 Comment 1:
The article “"Anees has Measles": Storytelling and Singing to Enhance MMR Vaccination in Child Care Centers Amid Religious Hesitancy” has been reviewed . This is a community intervention to reach out hesitant parents that are not prone to vaccinate their children against measles.
The study considers how an entertainment narrative about childhood vaccination influences related attitudes and positive issue-related thinking.
A few considerations:
Introduction line 80
The aim of the study is to promote vaccine acceptance among healthcare workers and parents, I believe children do not have much to say about being vaccinated or not.
Author Response:
Thank you for your valuable feedback on our manuscript. We appreciate your thoughtful comments and suggestions for improvement. Regarding your point on the study's aim, we have revised the text accordingly. The updated sentence now reads: "These efforts focus on creating culturally resonant narratives and imagery to enhance health literacy and promote vaccine acceptance among child care center staffs and parents." This change reflects the focus on the key decision-makers in the vaccination process. We appreciate your attention to detail and believe this revision better aligns with the study's objectives.
Reviewer 3 Comment 2:
- Materials and Methods
2.1 Study Population: The study population consisted of all children aged 18 months to 5 years only? The intervention seems to focus on children but they really have no power to determine their immunization . Target population should be their parents, child care center’s staff , religious counselors etc, that should be instructed through storytelling to their children. If a district has a low uptake for MMR it isn’t because children refuse to get vaccinated.
In the context of an outbreak people become more concerned about the disease and therefore are more prone to protect their children. In fact the survey is carried out among these adult populations, so this should be metioned as study population in methods
Author Response:
Thank you for your insightful feedback on our manuscript. We appreciate your comments and agree that the target population for the intervention should include parents, child care center staff, and religious counselors, as they are the key decision-makers regarding children's immunization. We have revised the "Study Population" section to reflect this.
Revised Section 2.1:
2.1 Study Population
The study population included parents, child care center staffs, and religious counselors associated with children aged 18 months to 5 years in subdistrict local government child care centers across 8 districts with recorded measles mortality, located in 5 provinces of Deep South Thailand. The data collection period spanned from December 1, 2019, to March 2020. During this time, an immunization campaign was implemented, which included distributing a culturally adapted storybook and organizing measles prevention activities such as storytelling and singing at the child care centers. These districts were selected based on their historically low uptake of the MMR vaccine and recent reports of measles outbreaks, targeting areas where the intervention could have the most significant impact.
Reviewer 3 Comment 3:
What statistical analysis should be in the Methods section. Levels of significance , CI?
Author Response:
Thank you for your valuable feedback on our manuscript. We appreciate your suggestion to include more details on the statistical analysis in the Methods section. We have revised the Methods section to incorporate the levels of significance and confidence intervals (CI) used in our analysis.
Revised Methods Section:
Vaccination Uptake Analysis:
MMR vaccination coverage at 12 and 48 months was assessed based on records from the local health clinics and compared with broader regional and national data. The differences in vaccination rates before and after the intervention were analyzed using the Wilcoxon signed-rank test due to the non-normal distribution of the data (as indicated by the Shapiro-Wilk test). Statistical significance was determined at a level of P < 0.05. Confidence intervals (CI) for the changes in vaccination rates were calculated to provide a measure of the precision of the estimated effects.
Trend Analysis in Immunization Coverage:
The monthly administration of MMR doses during 2020 was tracked alongside significant outbreak events and compared with historical data from 2018 to 2019 to identify any significant changes in vaccination trends. The analysis involved comparing the pre-intervention and post-intervention periods, with statistical significance determined using appropriate non-parametric tests.
Reviewer 3 Comment 4:
- Results
Statistical assessment should include age, and sociodemographic features of parents that can affect uptake when compare to regional and national level
- coverage rate of 44.3% (435/983) before the intervention…. Was this also before the outbreak? How many children were affected in the outbreak? Were there any deaths?
Author Response:
Author Response:
Thank you for your insightful comments. We appreciate your suggestions and would like to address each point:
- Statistical Assessment Including Age and Sociodemographic Features:
Our study primarily focused on vaccination uptake, and we did not collect detailed information on the age and sociodemographic features of parents that could affect uptake when compared to regional and national levels. We acknowledge this as a limitation and have added it to the limitations section.
- Coverage Rate Before the Intervention:
The coverage rate of 44.3% (435/983) before the intervention was measured during the outbreak. Information regarding the number of children affected in the outbreak and mortality rate is provided in the introduction section.
Revised Limitations Section:
The study, while insightful, presents several limitations that should be considered when interpreting the results. Firstly, the absence of a control group limits the ability to conclusively attribute the observed improvements in vaccination rates solely to the intervention, despite comparisons with regional and national data. This limitation makes it challenging to definitively isolate the effect of the intervention from other external factors that may have influenced the outcomes. Secondly, the findings are based on a relatively small sample size and are confined to specific child care centers in Southern Thailand, which may not fully represent broader populations or different cultural contexts. Thirdly, the reliance on voluntary participation could introduce selection bias, as those who chose to participate might inherently have different attitudes towards vaccination compared to the general population. Additionally, we did not collect detailed information on the sociodemographic features of parents, which could provide further insights into factors affecting vaccination uptake. While the 48-month follow-up period provided valuable longitudinal data, the study could benefit from exploring other aspects such as the continuity of vaccination behavior beyond the immediate aftermath of the intervention and the long-term acceptance of vaccines in the community. These limitations underscore the need for further research with a more robust experimental design, including a control group and broader demographic sampling, to validate and expand upon these initial findings.
Revised Introduction Section:
Since 2000, measles cases in Thailand have generally trended downwards, but the Deep Southern provinces have experienced recurring epidemics every 3-5 years, leading to disability and mortality. These provinces, where Muslims are the majority, reported the highest incidence rates in the country, with 126 cases per 100,000 population, and notably low MMR vaccine coverage of around 70% from 2014 to 2018 [28-30]. The region's overall childhood vaccine coverage is among the lowest in Thailand, compounded by the highest frequency of measles outbreaks. In November 2018, more than 20 cases of measles-related deaths were reported for that year's outbreak, highlighting the severe impact of these epidemics [28-32]. In this area, vaccine acceptance is significantly influenced by Islamic practices and religious leaders, contrasting with other Thai communities, predominantly Buddhist, who generally perceive vaccines as protective against disease [33]. This pattern is similar to that observed in Malaysian Muslim communities, where vaccine hesitancy is also prevalent [34].
Round 2
Reviewer 3 Report
Comments and Suggestions for Authors
Thank you. Authors have satisfactorily made all corrections and adhered to suggested changes.